# Old Dumped Fly Ash as a Sand Replacement in Cement Composites

**Jolanta Harasymiuk * and Andrzej Rudziński**

Institute of Building Engineering, Faculty of Geodesy, Geospatial and Civil Engineering, University of Warmia and Mazury in Olsztyn, Heweliusza 4, 10-724 Olsztyn, Poland; andrzej.rudzinski@uwm.edu.pl

* Correspondence: jolanta.harasymiuk@uwm.edu.pl; Tel.: +48-89-523-4586

**Abstract:** The use of industrial residues to replace natural resources for the production of building materials is economically and ecologically justified. Fly ash (FA) taken directly from electro-filters is commonly used as a cement replacement material. This is not the case, however, for old dumped fly ash (ODFA) that has been accumulating in on-site waste dumps for decades and currently has no practical use. It causes environmental degradation, which is not fully controlled by the governments of developed countries. The aim of the study was to assess the possibility of using ODFA as a partial replacement for sand in cement composites. ODFA replaced part of the sand mass (20% and 30%) in composites with a limited amount of cement (a cement-saving measure) and sand (saving non-renewable raw material resources). ODFA was activated by the addition of different proportions of hydrated lime, the purposes of which was to trigger a pozzolanic reaction in ODFA. The quantitative composition of the samples was chosen in such a way as to ensure the maximum durability and longevity of composites with a limited amount of cement. The 28-day samples were exposed to seawater attack for 120 days. After this period, the compressive strength of each sample series was determined. The results suggest the possibility of using ODFA with hydrated lime to lay town district road foundations and bike paths of 3.5 to 5 MPA compressive strength. What is more, these composites can be used in very aggressive environments.

**Keywords:** old dumped fly ash (ODFA); sand replacement; lime; durability of cement composites with ODFA

## 1. Introduction

The 21st century is characterized by the desire to, potentially, fully implement sustainable growth in all areas of social and economic life. This trend also concerns the construction sector. Globally, in Europe and particularly in Poland, construction projects for enterprises that potentially significantly affect the environment always require an environmental impact assessment [1]. Moreover, a habitat assessment may precede the implementation of every construction project with a significant impact on the area or the areas incorporated in the ecological network Natura 2000 [2]. Both analyses of construction and material solutions for building structures, as well as environmental assessments of commercial buildings are common and are carried out in order to reduce their environmental impact [3–5]. However, in recent years, attitudes towards the design of building structures have changed. Presently, the industry should consider the effectiveness of solutions for the whole life spans of buildings [6,7]. Thus, the continuous search for new material solutions using industrial waste is necessary [8,9]. One example of this type of industrial waste is fly ash (FA), which is a dust collected in electro-filters after the combustion of hard or brown coal in the different combustion chambers of energy plants (e.g., power stations, heat and power plants). The overwhelming majority of FA is regained directly from production (FA from electro-filters). It is a valuable component of cements and

cement composites (mortars and concretes). However, a troublesome waste that spoils the natural environment is the FA that has for decades accumulated in on-site waste dumps, often close to urban areas. The widespread phenomenon of depositing by-products of coal combustion at factory dumps is associated with the dominant mode of conventional energy production. The main source of electricity and heat in the world is still the combustion of hard coal and lignite. Figure 1 shows that the amount of FA produced in 2010–2017 in Poland slightly decreased, whereas the amount of FA landfilled considerably increased in the same period of time [10–17].

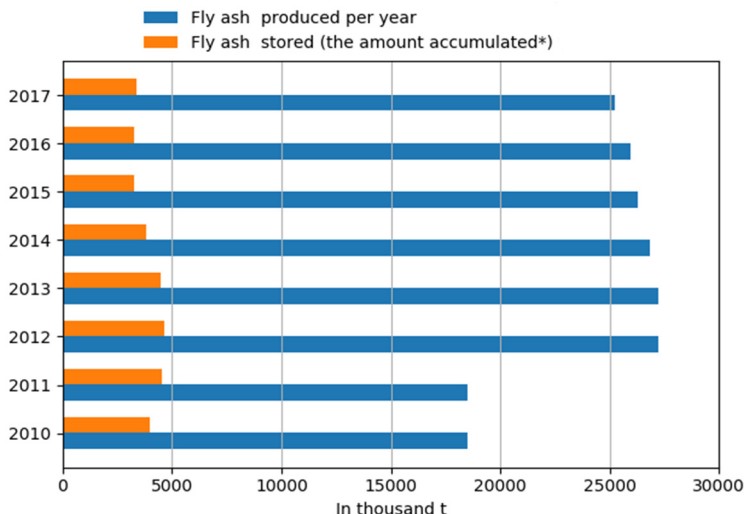

**Figure 1.** Graph of quantities of fly ash (FA) produced and deposited on landfills in Poland in 2010–2017.

This problem does not only apply to Poland [18,19]. As reported in [20], only a quarter of the total production of this waste is utilized. According to the Environmental Protection Agency (EPA), in the year 2008 FA was disposed of in over 310 active on-site landfills and 735 active on-site surface impoundments. Landfills averaged over 120 acres in size and 40 feet in depth, while impoundment areas averaged over 50 acres in size and 20 feet in depth. With the dominance of the current mode of conventional energy production and the poor tendency to utilize freshly produced FA, it can be assumed that the problem will not be reduced in the near future. Thus, new applications for ODFA, should be sought, which makes studies on this category of waste indispensable.

Scientific publications devoted to the use of ODFA in pastes, mortars or concretes are limited. Up until now, there have been many publications concerning the usage of FA taken directly from electro-filters. Such FA has been used mainly as a cement replacement. Only in recent years has the potential for the partial replacement of cement by FA from electro-filters in concretes (up to 50% of cement mass) been discussed [21–23]. According to the PN-EN 206+A1:2016 standard [24], the share of FA in mortars and concretes may amount to up to 30% of the cement mass, meaning that the use of this waste is in the range of 100–120 kg/m³. Our study develops the use of ODFA as a partial replacement for sand (20% and 30% of sand mass) in cement composites. It involves the use of this waste in the range of 300–400 kg/m³.

Due to its lack of binding capacity, FA requires the use of certain additives to serve as binding activators. In [25], sodium sulphate was used as a binding activator, whereas in [26], a paste with FA was activated with sodium hydroxide. In [27], calcium hydroxide made up 9.4% of the paste mass as an activator for a paste with FA. The amount of calcium used in our study ranged from 0.31% to 0.92% and from 0.46% to 1.31% of the composites' mass, respectively (the remaining part, made up of lime, came from the hydration of alite and belite in cement). These amounts depended on the assumed content of ODFA in the composites. Our previous study (unpublished) showed that this amount of calcium is optimal when considering the compressive strength of the standard sample (28 days) and the prolonged sample (148 days).

The use of any kind of FA to produce cement composites is reasonable only in the case of obtaining the assumed technological properties of such materials, and first of all, durability. The problem of durability of cement composites was the subject of numerous publications [28–31]. The reason for that is the high pollution level of the atmosphere and water, which increases the aggressiveness of the environment and the speed of corrosion. In the publications referred to, however, mainly the durability of concrete was dealt with. In our research, we dealt with the durability of composites with a considerable addition of ODFA (240% and 360% of cement mass). The most frequent aggressions are chlorides (the sources of chloride salts are defrosting agents and sea water) and sulfates. The influence of chloride salt activity on building mortars with FA corresponding to the PN-EN 450:2012 standard [32] was examined in [33,34]. Permeation of chloride to concrete was examined in [35]. The impact of sulfate ion activity on concretes was examined in [36–40]. Seawater is a more complex corrosive agent than the above-mentioned agents. It contains a series of salts (mainly chlorides and sulphates) that work simultaneously. In [41] the influence of various aggressive solutions, namely HCl, $Na_2SO_4$ and seawater, on cement pastes and mortars was monitored. The activity of seawater on cement pastes without addition of FA was examined in [42]. However, the concentration of the aggressive media in these publications was low. In our research, we dealt with the effect of seawater on technological properties of cement composites with a large amount of ODFA. Seawater had a salinity of 4.3%. According to [29], such salinity is high, more than two times higher in comparison to other studies [43]. The assumed concentration of seawater solution led to an intensified process of aggression.

Summing up, there are no publications devoted to the usage of ODFA as an ingredient of cement composites. There is also a lack of publications pertaining to the durability of such composites in very aggressive environments.

The aim of this study was to evaluate the possibility of using the highest amount of useless energy-producing waste material (ODFA) as a partial replacement of sand in cement composites. Using ODFA reduces degradation of the natural environment caused by storage of FA and can solve the problem that the governments of developed countries fail to manage.

The composites under study contained a small amount of cement (25% of the standard mortar mass) and a large proportion of ODFA (240% and 360% of the mass of the used cement). A partial replacement of sand with ODFA in cement composites would decrease the demand of the road building materials market for this non-renewable raw material. A reduction of cement content in composites is connected to a decrease in emissions of the $CO_2$ related to its production.

## 2. Materials and Methods

### 2.1. Materials

In this study, locally available building materials were used. Table 1 shows the quantitative and qualitative characteristics of these materials.

From the examined ODFA, natural sand of the grain size of 0–2 mm, Portland cement and calcium hydroxide, a series of sand–ash–cement composites were made, the qualitative and quantitative composition of which are shown in Table 2.

**Table 1.** Characteristics of the materials used in the research.

| Materials | Quantitative and Qualitative Characteristics |
|---|---|
| Natural sand with grain size of 0–2 mm obtained from Eco-Ter in Kronowo (Poland) gravel pit | The petrographic composition of sand was classified according to the PN-EN 932-3:1999 standard [44]. The sand contained 80.9% of quartz, chalcedony and opal, 12.5% of magma and metamorphic rocks and 6.6% of sedimentary rocks. The alkaline reactivity was 0. |
| ODFA obtained from Michelin S.A. Company (Poland) | The loss of ignition of ca. 15% classified according to the PN-EN 196-2:2013-11 standard [45], granulation containing ca. 20% of the up to 0.045 mm fraction, yet not containing unbound calcium. |
| CEM I-32, 5R Portland cement obtained from the Ożarów cement plant (Poland) | The Portland cement was classified according to the PN-EN 197-1:2012 standard [46]. |
| Hydrated lime obtained from Natura production facility (Poland) | The lime was classified according to the PN-EN 459-1:2015-06 standard [47]. Its content was CaO + MgO 95.2%, MgO 0.7%, $CO_2$ 1.8%, $Ca(OH)_2$ 91.4%, $SO_3$ 0.1%, $H_2O$ 0.8%. |
| Seawater solution | Composition of seawater solution consisted of (in 1 $dm^3$) 30.1 g NaCl, 6.0 g $MgCl_2$, 5.0 g $MgSO_4$, 1.5 g $CaSO_4$, 0.2 g $KHCO_3$ (p.a.), according to [29]. |
| Distilled water | - |

**Table 2.** The qualitative and quantitative composition of the composites.

| Series | Cement (kg/m³) | Sand (kg/m³) | ODFA (kg/m³) | Water (kg/m³) | Ca(OH)₂ (kg/m³) |
|---|---|---|---|---|---|
| 0 * | 136.3 | 1639.0 | - | 267.1 | - |
| $2H_0$ | 132.9 | 1288.4 | 319.1 | 307.3 | - |
| $2H_2$ | 130.2 | 1250.2 | 312.5 | 306.8 | 6.3 |
| $2H_4$ | 124.7 | 1196.9 | 299.2 | 299.2 | 12.0 |
| $2H_6$ | 120.3 | 1155.6 | 288.9 | 294.3 | 17.3 |
| $3H_0$ | 132.3 | 1111.8 | 476.4 | 329.4 | - |
| $3H_2$ | 129.0 | 1083.9 | 464.5 | 326.9 | 9.3 |
| $3H_4$ | 127.1 | 1067.9 | 457.7 | 327.7 | 18.3 |
| $3H_6$ | 123.0 | 1065.7 | 442.7 | 322.4 | 26.6 |

\* Control sample.

　　　As the "0" series, the qualitative composition close to the standard mortar was assumed (used to determine cement classes), but the standard sand was replaced with natural sand and the mass of cement was decreased by 75% (from 450 g according to PN-EN 196-1:2006 [48] to 112.5 g). In the composites, 20% and 30% of the mass of sand was replaced by ODFA.

　　　The composites were activated by the addition of hydrolyzed lime in the amount of 0%, 2%, 4% and 6% of the mass of the added ODFA (series "$2H_0$", "$2H_2$", "$2H_4$", "$2H_6$", "$3H_0$", "$3H_2$", "$3H_4$", "$3H_6$") (Activation of FA can be carried out with two methods, mechanical and chemical. The mechanical method consists in the fragmentation of FA [49] and is expensive. Chemical activation is achieved by the addition of activators to the composite, e.g. calcium hydroxide [50], sodium sulphate [25] and sodium hydroxide [26]. Chemical activation using calcium hydroxide was applied in the work.) This was done because ODFA is a Pozzolan that has no binding properties but rather reacts with Ca $(OH)_2$ formed during the hydration of alite and belite from cement. With a limited amount of cement in composites (from 124.7 to 132.9 kg/m³), an insufficient amount of Ca $(OH)_2$ is formed (assuming that the cement contained about 55% alite and about 12% belite). The insufficient amount of $Ca(OH)_2$ in the composites had to be replenished.

## 2.2. Sample Preparation

The samples of ODFA were taken from the heap located nearby the Michelin-Polska factory in Olsztyn (Poland), from a depth of ca. 1 m. The ingredients of fresh mixtures were blended in a laboratory concrete-mixer (according to the PN-EN 196-1:2006 standard [48]). The volume of water in individual series of these mixtures was determined experimentally to obtain a dense–plastic consistency. The consistency of fresh mixtures was examined on a flow table according to the PN-EN 1015-3:2000/A1:2005 standard [51]. The consistency was determined at a flow of 12 ± 0.5 cm. Then, cylindrical specimens were made with dimensions of 80 mm in diameter and 80 mm in height. All specimens were compacted in a Proctor device. After demolding, all specimens were transferred to the moisture chamber and kept at a relative humidity of 97% ± 3% and a temperature of 20 ± 2 °C, until their testing age (28 days). After this time, part of each specimen was left in the moisture chamber for 120 days. A second part was submerged in distilled water, and a third part was submerged in the seawater solution for 120 days.

## 2.3. Testing Methods

The assumed scope of the research consisted of both ODFA and hardened composites tests.

To determine the usefulness of ODFA as a partial sand replacement, conventional (grain size analysis, loss on ignition (LOI) determination) and modern research techniques (thermal analysis, scanning microscopy, electronic spectroscopy) were applied.

LOI determination was conducted according to the PN-EN 196-2:2013-11 standard [45]. Dried at 105 °C, the specimens were calcined at 950 °C. The percentage of organic matter (unburned carbon) was determined from the weight loss (the average value is presented in this work).

The test using the TGA–DTA thermal analysis of chemical processes in ODFA was made on a simultaneous TGA–DTA type SDT 2960 thermoanalyzer (TA Instruments). The thermogram was registered in atmospheric air with the speed of warming of 10 °C/min to 1000 °C. This test was made in the Chemical Analysis Laboratory of the Nicolaus Copernicus University in Toruń (Poland).

The test of ODFA structure was conducted with the help of scanning electronic microscopy (SEM). The observation was done with a JSM-5310LV scanning microscope.

The EDS analyses of ODFA were made with the aid of an SA Ranger spectrometer with a XFLASH detector.

The above-mentioned tests enabled physico-chemical characterization of ODFA.

Tests of hardened composites included density, weight water absorption and standard and long-term compressive strength.

Tests for density and water absorption were done for all the 28-day sample series.

The compressive strength of the composites with ODFA was determined after 28 days of setting and hardening, according to the PN-EN 14227-3:2004 standard [52].

To determine the durability of the composites, the 28-day samples were subjected to a very aggressive environment. They were submerged in a seawater solution for 120 days. For comparison, the successive sample series were submerged in distilled water and in a moisture chamber for 120 days. The results of the activity of the seawater solution and the distilled water on the composites were determined on the basis of a measurement of compressive strength according to the PN-EN 14227-3:2004 standard [52]. The structure of the samples after seawater attack was examined with the aid of a JSM-5310LV scanning microscope.

## 3. Results

## 3.1. Old Dumped Fly Ash Tests

The results of ODFA tests are shown in Figure 2, Figure 3, Figures 4 and 5a–c and Table 3.

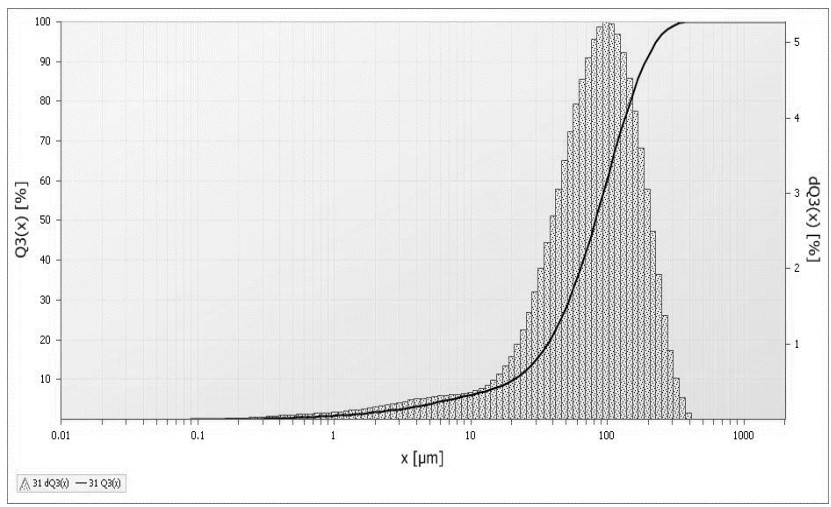

**Figure 2.** Curve of ODFA grain size.

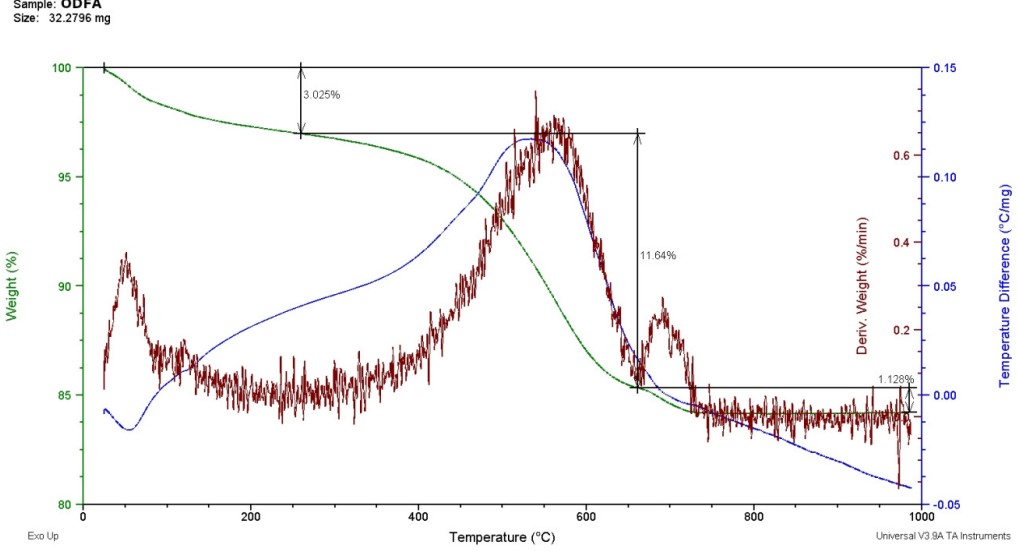

**Figure 3.** Thermal analysis—TG and DTA curves of ODFA; the green curve (TG) represents the change in mass of the sample in temperature function; the blue curve (TGA) represents the change of temperature of the sample in regard to the reference sample.

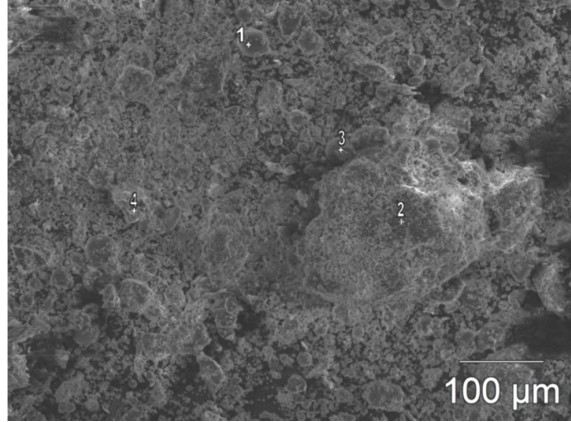

**Figure 4.** SEM image of ODFA; points 1–4 represent selected points for EDS analysis.

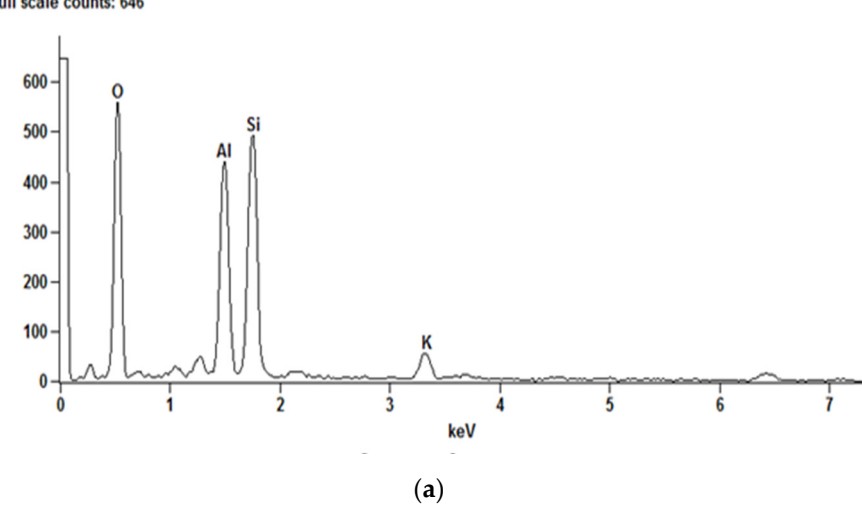

(a)

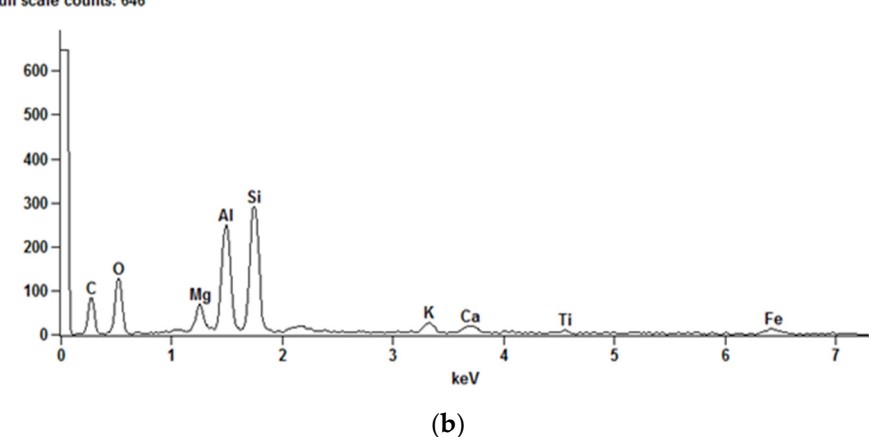

(b)

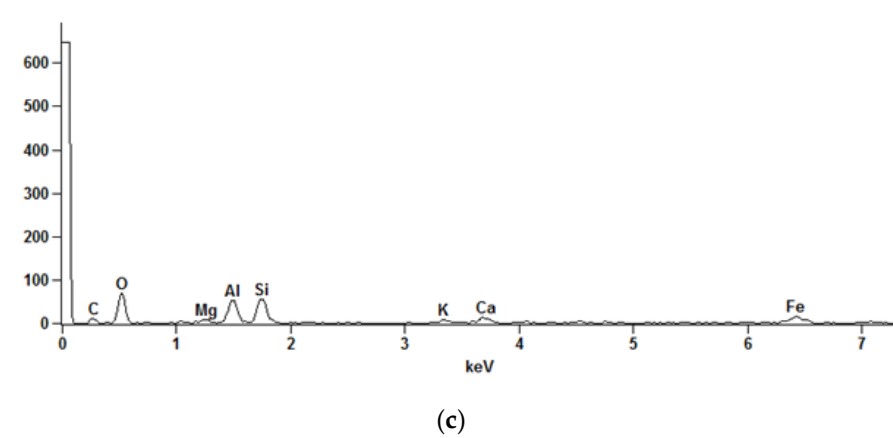

(c)

**Figure 5.** (**a**) EDS analysis of ODFA in the point 1 in Figure 4; (**b**) EDS analysis of ODFA in the point 2 in Figure 4; (**c**) EDS analysis of ODFA in the point 2 in Figure 4.

**Table 3.** The oxide composition of ODFA.

| Oxide | Content (%) |
|-------|-------------|
| $SiO_2$ | 40.46 |
| $Al_2O_3$ | 25.00 |
| $Fe_2O_3$ | 10.21 |
| CaO | 9.81 |
| MgO | 2.44 |
| $SO_3$ | 0.25 |
| $K_2O$ | 0.26 |
| $Na_2O$ | 0.49 |
| $P_2O_5$ | 0.21 |
| $TiO_2$ | 1.21 |
| MnO | 0.15 |
| SrO | 0.04 |

The fineness of FA had a considerable influence on its Pozzolanic activity [53]. The fractions below 45 μm were characterized by strong Pozzolanic activity. Figure 2 shows that the fractions up to 45 μm made up less than 20% in the examined ODFA. Hence, because of fineness, ODFA was not in accordance with the PN-EN 450-1:2012 standard [32].

A property of FA that is important from the practical point of view is LOI. The result of a high LOI is an enhanced water demand of FA (compared to FA of low calcination losses) [53]. According to the PN-EN 450-1:2012 standard [32], it should be 5%, while in our research it was 15%.

A supplement to the above-mentioned test was a more detailed TGA–DTA analysis (Figure 3).

As seen in Figure 3, ODFA contained 3.025% of $H_2O$ and 12.76% of organic matter, mainly of coal. Both mentioned tests showed that the content of unburnt coal in ODFA significantly surpassed the amount accepted in concretes.

The results of the morphological observation of ODFA are presented in Figure 4. A microscopic observation was made at a magnification of 200x. The SEM image showed that examined ODFA was fine-grained. The majority of the grains were spherical, with a smooth surface and a diameter up to 45 μm.

The EDS analyses of ODFA showed the presence of a number of elements in marked points (Figure 5a–c).

The maximum values were recorded for Si and Al. This corresponds to the content of their oxides ($SiO_2$ and $Al_2O_3$) given in Table 3.

A spectrographic analysis showed the qualitative and quantitative composition of ODFA. The presented results were average values from three tests. Based on the content of the three basic oxides ($SiO_2$, $Fe_2O_3$, $Al_2O_3$), it was determined that the ODFA examined was a siliceous FA (Table 3).

ODFA used in the research did not meet the requirements of the PN-EN 450-1:2012 standard [32] because of its losses on ignition and its fineness. However, there have been contradictory research results concerning the influence of high ignition losses on compressive strength of composites with FA [22].

*3.2. Cement Composite Tests*

3.2.1. Compressive Strength before and after Seawater Attack

All the specimens were examined for compressive strength after 28 and 148 days of setting and hardening. Table 4 presents the results of this test.

**Table 4.** Results of the compressive strength of the 28-day and 148-day samples as well as the 28-day samples after 120 days of submersion in distilled water and seawater.

| Series | Compressive Strength (MPa) | | | |
|---|---|---|---|---|
| | 28 Days in a Moisture Chamber | 28 + 120 Days in a Moisture Chamber | 28 + 120 Days in Distilled Water | 28 + 120 Days in Seawater |
| 0 | 1.85 | 1.95 | 2.02 | 1.68 |
| $2H_0$ | 1.90 | 2.20 | 2.20 | 2.05 |
| $2H_2$ | 3.62 | 4.90 | 3.50 | 3.07 |
| $2H_4$ | 4.21 | 5.40 | 3.65 | 3.12 |
| $2H_6$ | 4.34 | 5.25 | 3.44 | 3.85 |
| $3H_0$ | 2.00 | 2.20 | 2.25 | 2.20 |
| $3H_2$ | 3.90 | 5.72 | 3.06 | 3.50 |
| $3H_4$ | 5.00 | 6.20 | 3.56 | 3.72 |
| $3H_6$ | 5.07 | 5.90 | 3.78 | 3.63 |

Figure 6a,b show an error bar depicting the standard (28-day) and the long-term (148-day) compressive strength of the series of composites kept in a moisture chamber.

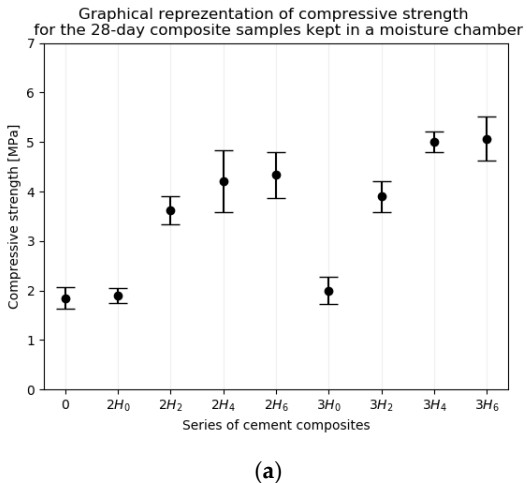

(**a**)

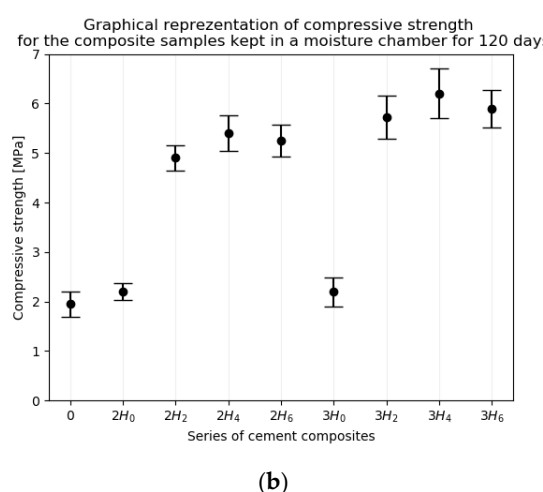

(**b**)

**Figure 6.** *Cont.*

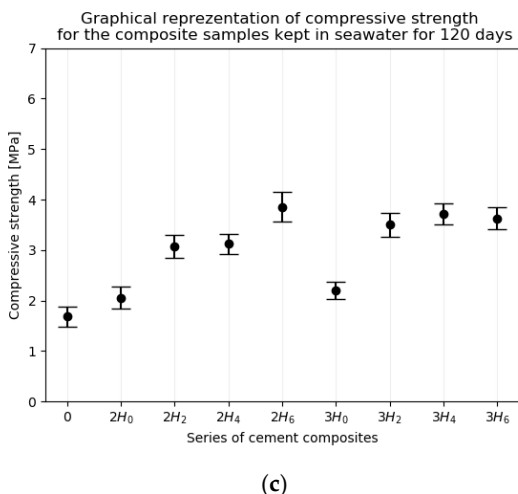

(**c**)

**Figure 6.** (**a**) Compressive strength data of the 28-day series of composites kept in a moisture chamber; (**b**) Compressive strength data of the 148-day series of composites kept in a moisture chamber; (**c**) Compressive strength data of the 148-day series of composites kept in seawater.

Figure 6c shows an error bar depicting the long-term (148-day) compressive strength of the series of composites kept in seawater solution. Each box in Figure 6a–c represents the compressive strength of 6 separate specimens. The dots correspond to the mean value; the lines show standard deviation.

Figure 7 shows that all the series of samples with ODFA and lime gained a higher compressive strength after 28 days in comparison to the control sample.

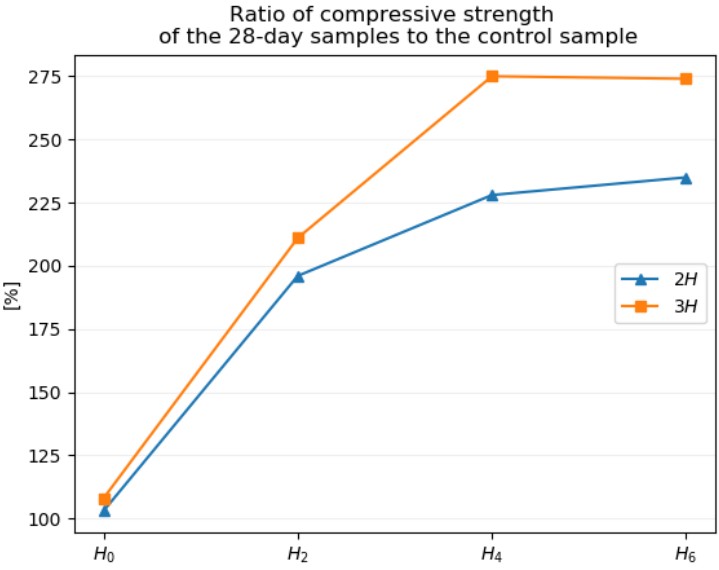

**Figure 7.** The changes in the compressive strength of the 28-day samples in comparison to the compressive strength of the control sample (the samples kept in a moisture chamber).

The maximum increase of compressive strength was observed in the "3H4" and "3H6" series of samples, which amounted to 270.2% and 274.1%, respectively.

The results of 120-day influence of the seawater solution on composites with ODFA and lime are shown in Figure 8.

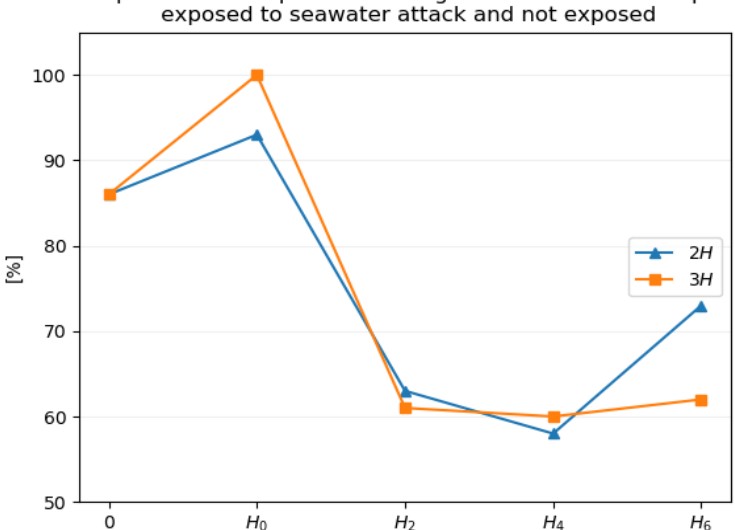

**Figure 8.** The changes of the compressive strength of the composite samples with ODFA and lime in comparison to the samples without lime subjected to the influence of seawater solution for 120 days.

These results were determined on the basis of the compressive strength measurement and were compared with the compressive strength of samples with the same composition but not subjected to seawater aggression. "2H0" and "3H0"series containing ODFA (but without lime) practically did not change their compressive strength. However, this strength was very low and amounted to about 2MPa (2.05 and 2.20 MPa). The strength of the "0" sample decreased by 13.8%. The compressive strength of the "3H0" series did not change, and "2H0" decreased by 6.8%. Series "2H2" and "2H4" and "3H2" and "3H4" obtained similar values (from 3.07 to 3.72 MPa). Their compressive strength dropped by 57.8% and 62.6%, respectively. Larger differences in decreases of compressive strength were noted for the "2H6" and "3H6" series, amounting to 61.5% and 73.3%, respectively.

### 3.2.2. Density

A density test is a basic examination conducted on all building materials. Figure 9 shows an error bar depicting the density of all the composites series after 28 days of binding and setting. Each box represents the density of four separate specimens.

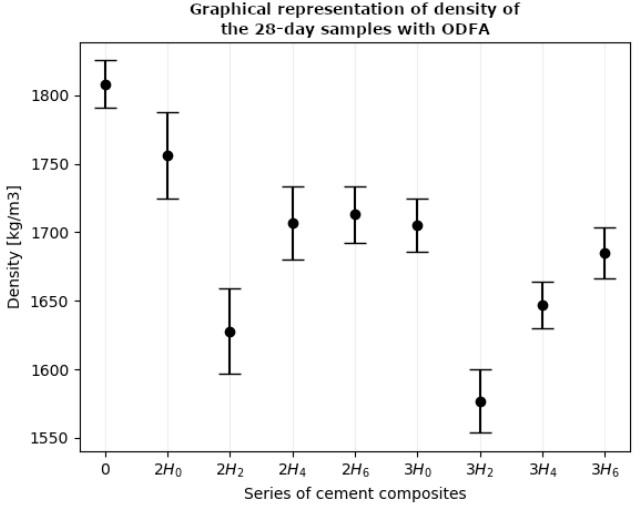

**Figure 9.** Density data of the 28-day series of composites.

Figure 10 shows the changes of the density measurements of all the composite series after 28 days of binding and setting.

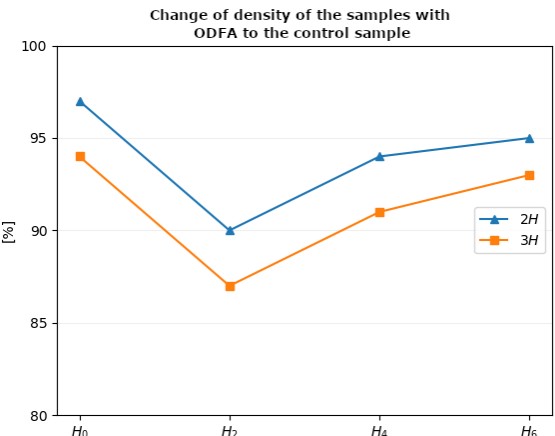

**Figure 10.** Increase in density of the composite samples with ODFA and lime in comparison to the control sample.

The density of composites with ODFA ranged from 1628 to 1756 kg/m$^3$ (samples "2H") and from 1647 to 1705 kg/m$^3$ (samples "3H"). The "0" sample without ODFA had a density of 1808 kg/m$^3$. Generally, the "3H" samples had a slightly lower density than the "2H" samples.

### 3.2.3. Water Absorption

Weight of water absorption was determined by the ratio of the weight of absorbed water to the weight of the specimen in the dry state. Some standards require that the water absorption of concrete be limited up to a certain level (4%). This is often a difficult condition to be obtained in industrial production, not only for concretes [54].

It is known that the absorption of cement paste (assuming nonabsorbability of aggregate) depends on the amount of cement, W/C ratio and degree of cement hydration (maturation time and conditions). Figure 11 shows an error bar depicting the weight of water absorption of all the composite series after 28 days of binding and setting. Each box represents the density of 4 separate specimens. The results of determination of weight of water absorption of the examined composites are presented in Figure 12.

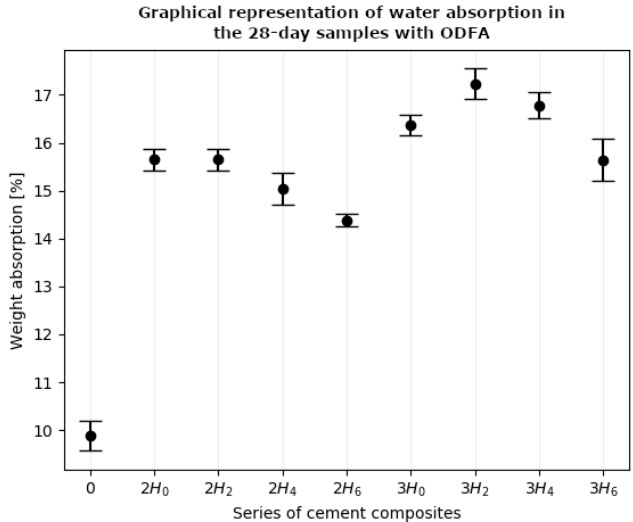

**Figure 11.** Water absorption data of the 28-day series of composites.

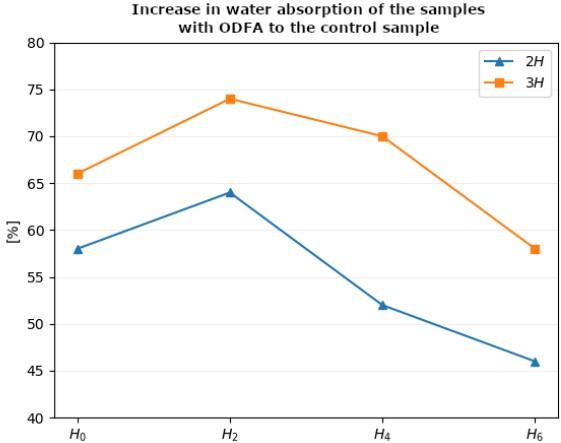

**Figure 12.** Increase in water absorption of the 28-day series of composites with ODFA and lime in comparison to the control sample.

The water absorption value of the samples with ODFA was considerably higher than the water absorption value of the control sample without lime and ODFA. For the "2H" samples the increase was from 44.1% to 63.8%. For the "3H" samples, the increase was still higher and ranged from 58.2% to 74.5%.

## 4. Discussions

The ODFA used in the research did not meet the requirements of the PN-EN 450-1:2012 standard [32] because of fineness and LOI. Despite this fact, the use of it in composites with reduced quantities of cement gave advantageous results. With a very limited quantity of cement in the hydration reaction, there was an insufficient amount of calcium hydroxide for a Pozzolanic reaction to occur. The applied ODFA contained virtually no unbound calcium oxide. The ODFA content in all the examined series of the samples considerably exceeded the cement content. To make a Pozzolanic reaction of ODFA possible, the additive calcium hydroxide in the amount of 0%, 2%, 4%, or 6% of the mass of ODFA was supplied to each of the series of samples.

Considerable increases in the compressive strength of composites with ODFA were obtained for the 148-day samples as compared to the 28-day samples. In the samples with ODFA and lime, the increase of compressive strength, as compared to the strength of the control sample only with ODFA, went up from 116.4% to 146.7%.

The presence of a large amount of ODFA reduced the density of composites from 6.7% to 12.8%.

The high increase in water absorption was determined by the presence of a large amount of ODFA in the composites [53]. ODFA bound water more intensively than the sand it replaced.

The seawater solution applied in the tests contains salts, which most often are harmful to building materials. The chloride salts, which are a significant ingredient of seawater, can act adversely on the cement matrix. The deterioration of technological properties of cement-based composites and composites with ODFA due to aggressive activity of seawater is connected with a transfer of reacting ions to the place of reaction, i.e., to the interior of a material. It can occur as a result of two mechanisms:

-    migration and influx to the interior of a composite resulting from capillary pull [55],
-    ion diffusion resulting from the strength gradient in its different areas [56].

The first mechanism gives a much higher speed of transport, which triggers a quick aggression process. The role of large capillary pores in the acceleration of aggression is obvious. The coefficient of diffusion D decreases, and the solution strength of salts C increases with time, which is caused by the progressive hydration of cement [57]. While assessing the process of migration in a hardened cement paste, and probably also in an ash–cement paste, it should be borne in mind that this is

not pure diffusion. Diffusing ions react with the phases of the paste and are subject to absorption on the amorphous surface of the C–S–H phase. Chloride ions also react with hydrated calcium aluminate, giving Friedl's salt $C_3ACaCl_210H_2O$. This compound does not cause expansion, but on the contrary, it decreases the amount of expansive compounds created [28,42,56,58]. Chloride aggression to which composites with ODFA were exposed diminished their compressive strength. Simultaneous action on the mixtures of calcium and magnesium chloride salts, as well as magnesium sulphate, caused the decrease of compressive strength in all the series of the samples with ODFA and lime. However, the obtained compressive strength of the samples with ODFA and lime after seawater attack was definitely higher, by 49.8% to 87.8%, than the compressive strength of the "2H$_0$" and "3H$_0$" samples without lime. The composites with ODFA and lime, under the influence of seawater attack, created more distinct and compact crystals (Figures 13b and 14b), than the samples with ODFA but without lime (Figures 13a and 14a).

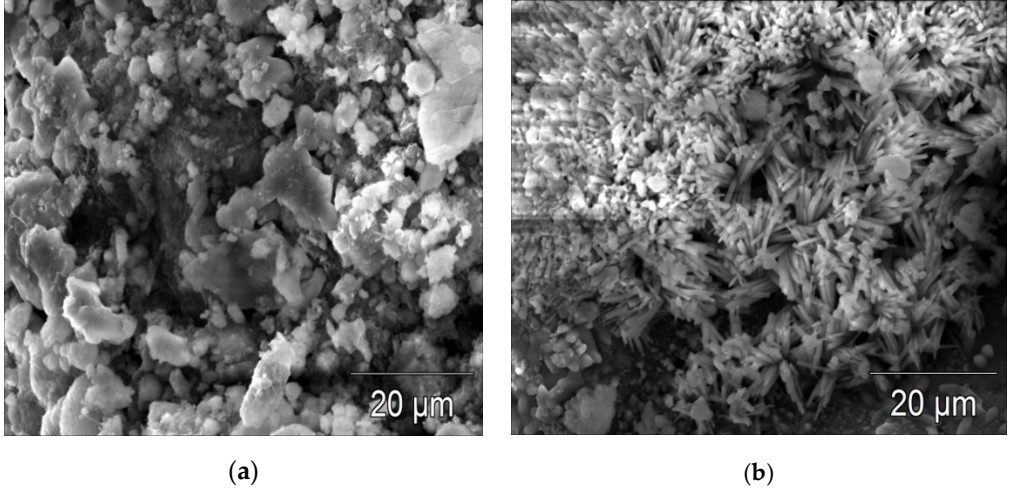

(**a**)            (**b**)

**Figure 13.** (**a**) SEM image of the 2H0 sample after seawater attack; (**b**) SEM image of the 2H6 sample after seawater attack.

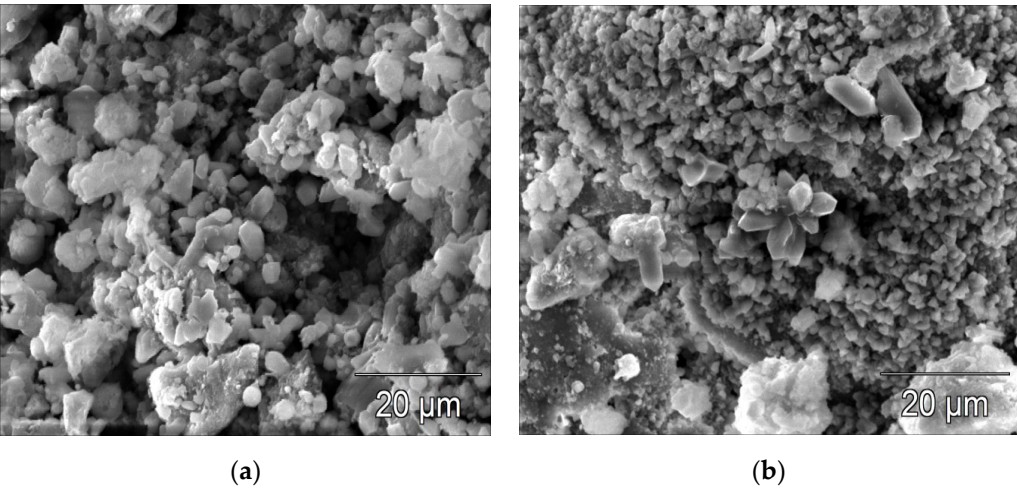

(**a**)            (**b**)

**Figure 14.** (**a**) SEM image of the 3H0 sample after seawater attack; (**b**) SEM image of the 3H6 sample after seawater attack.

These differences in the structure of the samples probably influenced the significant differences in these samples' compressive strengths. In terms of the composition of the seawater solution, beside the principal salt, NaCl, it also contained $MgCl_2$ and $MgSO_4$ magnesium salts, which caused not only a sulphate aggression, but also a magnesium one.

In the penetration of the composites with ODFA by the solutions containing sulphate ions, a significant role is played by the size of pores and the content of aluminate oxide and calcium oxide. The products of the reaction of magnesium sulphate with calcium hydroxide are, successively, calcium sulphate and hydrate, which in the presence of $C_3A$ turn into ettringite. Moreover, brucite can also be created. Generally, composites made of cement and FA are characterized by increased resistance to aggressive factors [58]. These properties, together with the increment of the FA content, strengthen the composite. In the case of sulphate aggression, its intensity decreases with the diminishment of $C_3A$ [28,59–63].

In spite of the very aggressive environment to which the samples of composites with ODFA, lime and a limited cement quantity were subjected to for 120 days, compressive strength of a minimum of 3 MPa was obtained.

## 5. Conclusions

The results of the research allow us to conclude the following:

1.  Large amounts of ODFA and a limited amount of cement were used in composites. The quantity of ODFA was, respectively, 240% and 360% of the cement mass used. Such volumes of this specific kind of FA are not used for the production of cement composites. This problem has also been virtually unaddressed in the research.
2.  With the usage of ODFA as a partial replacement of sand (20%–30%), it is possible to obtain the assumed technological properties of cement composites and limit the onerousness of this unwanted waste for the environment to a great degree. This is reasonable, but only with the addition of $Ca(OH)_2$.
3.  Composites with additives of different quantities of $Ca(OH)_2$ (2%, 4%, 6%) had considerably higher compressive strength than composites with the same quantity of cement and ODFA but without $Ca(OH)_2$.
4.  The samples with ODFA, but without $Ca(OH)_2$, obtained significantly lower compressive strengths (on average by ca. 2 MPa), both before and after the attack, compared to the samples with ODFA and $Ca(OH)_2$.
5.  The samples with ODFA and $Ca(OH)_2$ reached the compressive strength of 4.9–6.2 MPa before seawater attack and 3.07–3.85 MPa after it.
6.  Although ODFA did not meet the requirements of the PN-EN 450-1 standard pertaining to FA for production of concrete, their presence did not lead to a higher destruction in aggressive solutions than in composites made from traditional raw materials.
7.  Ash concretes are applied in road building and for construction of foundations with compressive strengths from 1.5 to 8 MPa. Composites with ODFA and $Ca(OH)_2$ as an activator can be applied for construction of town district road foundations and bike paths of the compressive strength class C3/4. Moreover, they can be used for foundations subjected to application of de-icing agents.

**Author Contributions:** The individual contribution and responsibilities of the authors were as follows: conceptualization, J.H. and A.R.; methodology, J.H. and A.R.; investigation, J.H. and A.R.; formal analysis, J.H.; writing—original draft preparation, A.R.; writing—review and editing, J.H.; visualization, J.H. and A.R. All authors have read and agreed to the published version of the manuscript.

**Funding:** This research received no external funding.

**Conflicts of Interest:** The authors declare no conflict of interest.

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
