# Peer review of "Old Dumped Fly Ash as a Sand Replacement in Cement Composites"

_buildings, doi:10.3390/buildings10040067_

Round 1

Reviewer 1 Report

Studies on the applications of ODFA are important, thus the study is interesting, but:

There are 14 Figures mentioned in the manuscript, but NO FIGURES included there. Thus the results are hardly to be evaluated.

English needs revision. There are spaces between words missing in several cases. Sometimes other mistakes occur.

Author Response

We would like to take this opportunity to thank the Reviewer for a review of the article entitled ”Old Dumped Fly Ash as a Sand Replacement in Cement Composites”(ID: buildings - 695727).We are thoroughly convinced that the comments contributed to improvement of our work.

We respond to the comments of the Reviewer in the order they are formulated:

1.     The whole article has been corrected for elimination of grammar mistakes.

2.     All figures were included in the text of the article.

We cordially thank for preparing the review.

Reviewer 2 Report

The manuscript targets an important topic that can reduce the use of traditional cement and thus further reduce the carbon footprint of the construction sector. It is useful to conduct a variety of testing techniques to support the main arguments of the manuscript. 

However, I could not find any figures in the manuscript. Therefore, I can not judge if the experimental results are able to support conclusions. I do suggest that figures must be included when the manuscript is resubmitted. 

Author Response

We would like to take this opportunity to thank the Reviewer for a review of the manuscript entitled ”Old Dumped Fly Ash as a Sand Replacement in Cement Composites” (ID: buildings - 695727).

Replying to the comment of the Reviewer we confirm that all figures were included in the text of the manuscript.

We cordially thank for the work done in preparation of this review.

Round 2

Reviewer 1 Report

Some minor changes required at the following rows:

57: FA ash

181: Alone line below the figure 3

324: minimum 3

Author Response

We respond to the comments of the Reviewer in the order they are formulated:

The whole article has been corrected for elimination of grammar mistakes. 

In line 57, the word ”fly ash” was replaced by its abbreviation – FA.

In the corrected version of the article, the lonely line below Figure 3 was deleted.

In line 324 a space between words ”minimum” and ”3 MPa” was made.

Reviewer 2 Report

The manuscript mentions many times about corrosion, such as chloride corrosion, corrosion of composition, etc. This is confusing. Corrosion is a process that a metal is transformed into a more stable state, such as its oxides. For concrete, if you mention corrosion, it generally means the corrosion of steel. Since there is no steel in the specimens, I would not suggest to talk about corrosion.

In Table 1, about seawater composition, were these chemicals added in 1 l water?

Line 155, the authors should explicitly say that the detector is used for EDS.

Table 3, the EDS only provides the possible ratio of elements. How did the authors determine the quantity of these chemicals?

Captions of figure 6 are a bit confusing.

Table 4 shows that storing specimens in either distilled water or seawater can lower the strength. I did not say the explanation for the case of distilled water.

The explanation for seawater is not so convincing. The authors mention sulphate, but do you think the concentration of sulphate is high enough to cause a problem?

The authors also mention the formation of ettringite, but I could not see any ettringite in SEM images. It is better to provide more evidence to support the arguments.

About the “corrosion resistance”, how was it measured? Are there any results shown in the manuscript?

Line 261, ROI should be LOI?

Round 3

Reviewer 2 Report

The authors claimed that the word ”corrosion” in the line was replaced by the word ”attack”. I do not see this change in the revised manuscript. 

I suggest using "EDS" instead of "microanalysis". The latter is so general. It is better to say the specific technique. 

The authors argued that the seawater solution can reduce the compressive strength due to the similar reactions with sulphate attack. But the authors should notice that in reality the sulphate concentration is not so high and its diffusion is slow, so this effect has a minor effect on the mechanical property of concrete.  

Author Response

The word ”corrosion” was replaced by the word ”attack” in the revised manuscript.

The captions under Figures 5a, 5b and 5c use the term ”EDS” instead of "microanalysis" in the revised manuscript.

The aggression caused by the seawater attack is a mixed chloride-sulfate aggression with a significant share of magnesium salts. In the study, in 1 dm3 seawater solution, the authors used 6g MgCl2 and 5g MgSO4. The results of research in the field regarding the seawater attack are ambiguous. The presence of NaCl in a solution containing sulfate ions does not weaken sulfate aggression, and even increases it, as it was demonstrated in the study. The aggressive effect of sulfate ions is smaller if there is NaHCO3 in the solution. The seawater solution used in our research did not contain NaHCO3, but only a minimal (trace) amount of KHCO3 (0.2g/dm3).